# A biallelic variant in *GORASP1* causes a novel Golgipathy with glycosylation and mitotic defects

Sophie Lebon[1], Arnaud Bruneel[2,3], Séverine Drunat[1,4], Alexandra Albert[1], Zsolt Csaba[1], Monique Elmaleh[5], Alexandra Ntorkou[5], Yann Ténier[6], François Fenaille[6], Pierre Gressens[1], Sandrine Passemard[1,7], Odile Boespflug-Tanguy[1,7], Imen Dorboz[1,7], Vincent El Ghouzzi[1]

GRASP65 is a Golgi-associated peripheral protein encoded by the *GORASP1* gene and required for Golgi cisternal stacking in vitro. A key role of GRASP65 in the regulation of cell division has also been suggested. However, depletion of GRASP65 in mice has little effect on the Golgi structure and the gene has not been associated with any human phenotype to date. Here, we report the identification of the first human pathogenic variant of *GORASP1* (c.1170_1171del; p.Asp390Glufs*18) in a patient combining a neurodevelopmental disorder with neurosensory, neuromuscular, and skeletal abnormalities. Functional analysis revealed that the variant leads to a total absence of GRASP65. The structure of the Golgi apparatus did not show fragmentation, but glycosylation anomalies such as hyposialylation were detected. Mitosis analyses revealed an excess of prometaphases and metaphases with polar chromosomes, suggesting a delay in the cell cycle. These phenotypes were recapitulated in RPE cells in which a similar mutation was introduced by CRISPR/Cas9. These results indicate that loss of GRASP65 in humans causes a novel Golgipathy associated with defects in glycosylation and mitotic progression.

## Introduction

*GORASP1* encodes the Golgi reassembly stacking protein GRASP65, a peripheral Golgi protein localized in the *cis*-Golgi and functioning as a membrane tether involved in Golgi ribbon formation in mammals (Ahat et al, 2019). Domains present in its N-terminal half are implicated in GRASP65 recruitment to Golgi membranes, notably via a myristoylation site and a direct interaction with the *cis*-Golgi matrix protein GM130 (Barr et al, 1998; Puthenveedu et al, 2006), and in a self-oligomerization process necessary for

membrane tethering (Rabouille & Linstedt, 2016). On the contrary, phosphorylation events in the C-terminal half prevent oligomerization of GRASP65 and favor Golgi ribbon unlinking at the onset of mitosis (Wang et al, 2003; Tang et al, 2012). GRASP55, which shares strong homology with GRASP65 in the N-terminal domain and is encoded by the *GORASP2* gene (Shorter et al, 1999), would play an analogous role but at the *medial/trans*-Golgi level (Rabouille & Linstedt, 2016). Given the importance of Golgi integrity in cellular homeostasis, numerous studies have attempted to clarify the role of GRASPs alone or in combination with membrane trafficking, protein glycosylation, or Golgi ribbon stacking or unlinking at the onset of mitosis, sometimes with different conclusions depending on the models used or the way the factors were down-regulated (Ahat et al, 2019; Zhang & Wang, 2020). So far, however, neither GRASP has been associated with pathology in humans. In mice, depletion of GRASP65 does not seem to induce a major phenotype in the tissues examined nor in the overall structure of the Golgi but leads to ribbon unlinking on the *cis* side, as well as N-glycosylation defects (Veenendaal et al, 2014). Glycosylation of plasma membrane proteins was also found impaired in HeLa cells depleted of GRASP65 by CRISPR/Cas9 (Bekier et al, 2017), whereas siRNA-based depletion caused defects in spindle dynamics and delayed cell cycle progression (Sütterlin et al, 2005; Tang et al, 2010). Herein, we report the identification of the first human *GORASP1* variant in a male patient with a neurodevelopmental disorder associating neurosensory, neuromuscular, and skeletal abnormalities. Molecular analyses show that the variant results in a stable transcript but a total absence of the GRASP65 protein. The overall structure of the Golgi in the skin fibroblasts from the patient was apparently normal, but the variant was associated with a decrease in terminal sialylation, as well as an increased mitotic index and a delayed prometaphase/metaphase transition. Mimicking the variant in RPE cells using CRISPR/Cas9 confirmed that these phenotypes are linked to the loss of GRASP65.

---

[1]Université Paris Cité, NeuroDiderot, Inserm UMR1141, Paris, France   [2]Université Paris-Saclay, Inserm UMR1193, Faculté de Pharmacie, Orsay, France   [3]AP-HP Département de Biochimie Métabolique et Cellulaire, Hôpital Bichat, Paris, France   [4]AP-HP Département de Génétique, Hôpital Robert Debré, Paris, France   [5]AP-HP Département de Radiologie Pédiatrique, Hôpital Robert Debré, Paris, France   [6]Université Paris Saclay, CEA, INRAE, Département Médicaments et Technologies pour la Santé, MetaboHUB, Gif sur Yvette, France   [7]AP-HP Département de Neurologie Pédiatrique, Hôpital Robert Debré, Paris, France

Correspondence: vincent.elghouzzi@inserm.fr; imen.dorboz@aphp.fr

# Results

## The patient's phenotype combines neurodevelopmental deficits with white matter abnormalities, neurosensory defects leading to deafness and myopia with vitreoretinal degeneration, neuromuscular contractures, truncal obesity, and skeletal abnormalities

For a detailed description of the patient's phenotype and follow-up, please see extended patient descriptions in Supplemental Data 1. The patient, aged 19 yr and 6 mo, is the third child of first cousin parents. No antenatal or neonatal events were reported, and developmental milestones for the first 2 yr of life were normal with independent walking acquired at 14 mo.

A progressive sensorineural deafness was detected at the age of 4 yr because of an absence of language and increased between the ages of 6 and 11 yr to −80 dB. At the age of 5, the patient underwent a brain MRI scan, which showed a normal appearance of the inner ear structures but revealed an abnormal white matter signal. The patient was then referred to and followed up by a child neurology department (LEUKOFRANCE).

Learning disabilities became obvious at elementary school, with difficulties with reading, writing, and counting. Hyperactivity with attention deficit was successfully treated with methylphenidate until the age of 14. Cognitive assessments confirmed poor verbal performance with difficulties in fluid reasoning and working memory. Writing and language tests specific to deaf children confirmed severe dyslexia. His head circumference curve has always been within the norm (50 p). A broad-based gait and slight instability with eyes closed were the only neurological signs observed during the course of the disease, in the absence of pyramidal and cerebellar dysfunctions.

White matter abnormalities revealed on initial MRI were diffuse, almost symmetrical, heterogeneous T2-weighted, and FLAIR (fluid-attenuated inversion recovery) hyperintensities in the frontoparietal white matter, with predominance in the periventricular zone (Fig 1A). Two bilateral cystic lesions (3–4 mm in size) were found in the body of the caudate nuclei, in contact with the walls of the lateral ventricle (Fig 1B). Subsequent MRI scans at 6.5 and 9.5 yr (data not shown) and at 14.5 yr (Fig 1C) showed a significant reduction in the extent and intensity of the white matter lesions, and the periventricular cysts remained stable. Sagittal T1-weighted MRI showed a normal aspect of the posterior fossa including the cerebellum (Fig 1D).

A thorough metabolic workup at 6 yr was normal (see details in Supplemental Data 1). Muscle pain was present at the age of 6, associated with stiffness in the leg muscles and frequent falls. Gabapentin and L-carnitine significantly improved the symptoms. Falls and morning stiffness became rare after the age of 10. Episodes of exercise intolerance occurred when he started playing handball after the age of 15, mainly in the form of cramps in the thighs and calves a few hours after exercise, but also occasionally in the form of acute knots in the vastus lateralis. The muscles appeared hypertrophic and stiff on contraction, particularly in the calves and biceps on the right side in a left-dominant patient. Muscle tests were always normal, and gait parameters improved over time from 10 to 19 yr. Serum creatine kinase and myoglobinuria

were always normal. EMG and nerve conduction velocities at ages 6 and 14 were normal.

Severe progressive myopia (−7 diopters) was discovered at the age of 12. 6 mo later, a systematic fundus examination revealed signs of lattice degeneration of the retina associated with vitreous condensation. Laser therapy was performed. At the last examination (18 yr and 7 mo), bilateral vitreous opacities had increased but retina and myopia remained stable.

Obesity (mainly truncal) developed after 5 yr, reaching a body mass index of 30 at 10 yr. Improvement was observed after 15 yr, and the normal body mass index limit (25) was reached at 18 yr.

Mild dysmorphic facial features and skeletal defects were also noted: an elongated face with micrognathia, a prominent nose, and downward-sloping palpebral fissures (Fig S1A and B). Skeletal X-rays revealed coxa valga (Fig S1C). Cervicodorsal kyphosis was associated with vertebral abnormalities compatible with Scheuermann's disease on MRI of the spine (Fig S1D). Ulna valgus and a genu valgum with recurvatum of the knees were also noted. Finally, bone thinning of the medial part of the distal epiphysis of the radius was observed (Fig S1E).

## Identification of a biallelic variant in *GORASP1*

Whole-genome trio sequencing identified four rare variants, homozygous in the patient. After analysis of these variants on the basis of their minor allele frequency, segregation with phenotype, computational predictions of pathogenicity, expression in relevant tissues, and involvement in disease-related pathways (see Supplemental Data 1 for details), the only relevant candidate was a homozygous variant in the *GORASP1* gene: NM_031899.4: c.1170_1171del p.(Asp390Glufs*18). Segregation of the *GORASP1* variant was confirmed within the family. The healthy parents, sister, and brother were all heterozygous for the variant (Fig 1E). *GORASP1* encodes the Golgi reassembly stacking protein GRASP65, a ubiquitously expressed peripheral Golgi protein involved in maintaining the structure and function of the Golgi apparatus. The identified variant was absent in the homozygous state in gnomAD (gnomAD v4.1), and observed in eight European (non-Finnish) heterozygotes (minor allele frequency = 0.000004956). It is expected to cause a frameshift from the aspartic acid codon 390, changing this amino acid to a glutamic acid residue and creating a premature termination codon at position 18 of the new reading frame p.(Asp390Glufs*18). The variant affects the C-terminal region of the serine/proline-rich (SPR) domain of GRASP65 (Fig 1F). To date, there are no known disorders associated with *GORASP1*. However, as several other Golgi proteins are associated with neurodevelopmental disorders with symptoms that are similar to our patient's phenotype (Rasika et al, 2018), we considered the *GORASP1* variant as a relevant causal candidate for the patient's disease.

## The variant in *GORASP1* results in a complete absence of the GRASP65 protein

To determine the effect of the patient's variant on *GORASP1* gene products, we first quantified mRNA expression in primary fibroblasts by quantitative PCR, using two pairs of primers distributed throughout the *GORASP1* transcript. No significant difference was

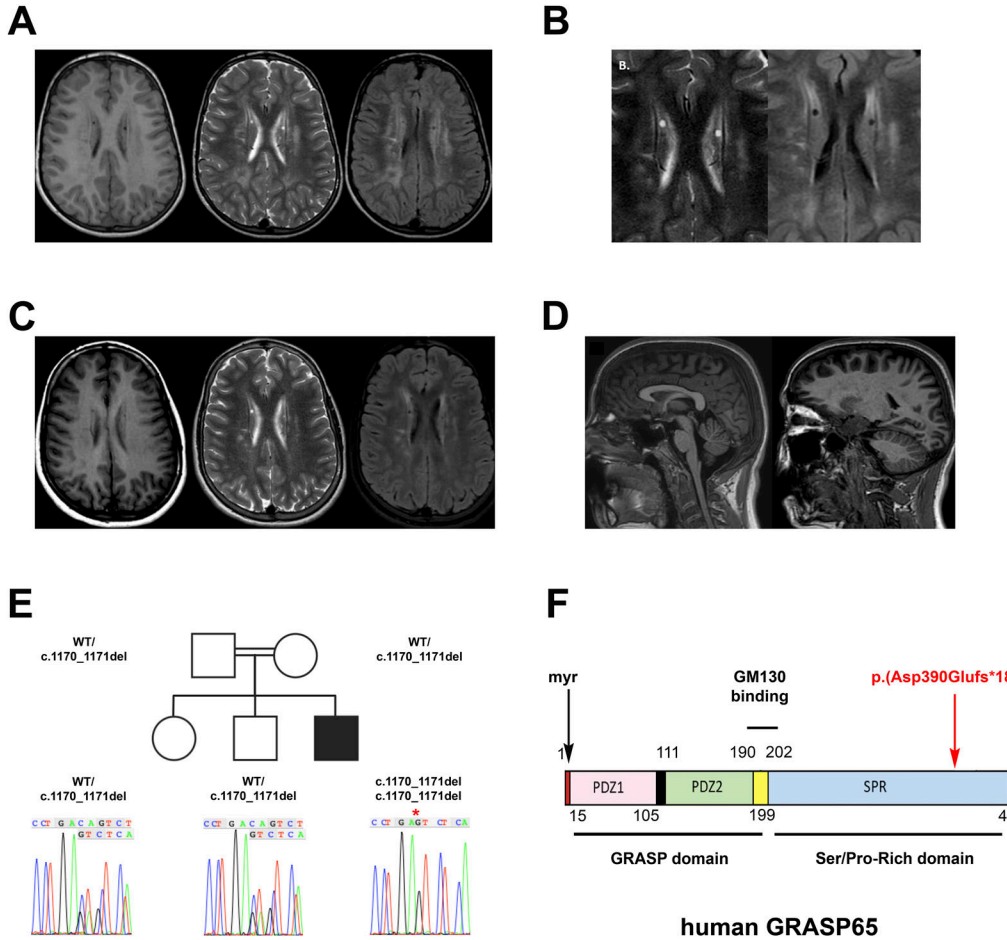

**Figure 1. Neurological features of the patient, genetic tree, and schematic representation of the human GRASP65.**
**(A, B)** Unenhanced T1/T2-weighted and FLAIR transverse images through the bodies of the lateral ventricles on initial MRI at age 5 showing severe, bilateral, and relatively symmetrical confluent hyperintensities in the periventricular white matter with preservation of the U-fibers in subcortical areas (A) and two tiny bilateral periventricular cysts unchanged during follow-up (B). **(C)** Unenhanced T1/T2-weighted and FLAIR transverse images through the lateral ventricle bodies at age 14, showing progressive reduction of bilateral periventricular hyperintensities and white matter. **(D)** Sagittal T2-weighted image showing normal appearance of the posterior fossa and cerebellum structures. **(E)** Family genetic tree and sequencing chromatograms showing segregation of the variant c.1170_1171del in the heterozygous state in healthy individuals and in the homozygous state in the patient. The red asterisk in the patient's chromatogram indicates the biallelic CA deletion. **(F)** Structure of the human GRASP65 showing the N-terminal half composed of the GRASP domain (AA15–199) with PDZ1 and PDZ2 subdomains, the GM130 binding domain (AA190–202), and the C-terminal half containing the serine/proline-rich domain (SPR, AA202–440), as well as the location of the identified variant (in red). myr: myristoylated glycine in the N terminus.

observed between patient and control fibroblast mRNAs in the expression level of the different amplicons, indicating that the variant does not exert any particular effect at the transcriptional level (Fig S2B). Next, we examined the expression of the GRASP65 protein in the same cells by immunoblotting and immunocyto-chemistry, using two anti-GRASP65 antibodies, one directed against the entire protein (GRASP65 FL) and the other specifically recog-nizing the C-terminal part (GRASP65 C-ter). In immunoblotting, a single band with the expected size of 65–70 kD was detected in the protein extract of control fibroblasts with both antibodies. However, no signal was visible in the patient's fibroblasts, regardless of the antibody used, indicating the absence of GRASP65 in the patient's cells (Fig 2D). Consistent with these results, immunocytochemical staining in control fibroblasts displayed a localization of GRASP65 in the Golgi apparatus, in perfect colocalization with GM130, indicating the *cis*-Golgi distribution of the protein. However, no signal could be detected in the patient's cells, regardless of the antibody used (Fig 2A and C). Of note, neither the protein level nor the subcellular local-ization of GM130 and GRASP55 appeared to be affected by the ab-sence of GRASP65 in the patient's cells (Fig 2B and D). These data indicate that the mutation identified in the patient leads to a loss of function of GRASP65 because of the absence of the protein without impacting GM130 or GRASP55. To determine whether the absence of

GRASP65 affects the Golgi structure in patient cells, we next analyzed the morphology of the Golgi apparatus by quantifying the area stained by the GM130 signal by immunofluorescence microscopy. No significant differences were detected between control and patient's cells, suggesting that the absence of GRASP65 does not significantly impact the overall Golgi morphology in fibroblasts when assessed by fluorescence microscopy (Fig 2E).

**GRASP65 loss results in glycosylation defects in the patient**

To determine whether the loss-of-function variant of GRASP65 identified in the patient affects plasma membrane protein glyco-sylation, we stained the surface of primary fibroblasts with a fluo-rescently labeled WGA that binds sialic acid and N-acetylglucosamine. A significant reduction in signal intensity was observed in the patient's fibroblasts compared with the healthy control (Fig 3A). To exclude that this difference could result from variations between two unrelated cell lines, we electroporated a GRASP65-GFP con-struct in the patient's fibroblasts. An intensity of WGA membrane labeling comparable to that of the healthy control was observed in electroporated fibroblasts, suggesting that GRASP65-GFP over-expression was able to restore plasma membrane protein glyco-sylation (Fig 3A). To determine whether these abnormalities could

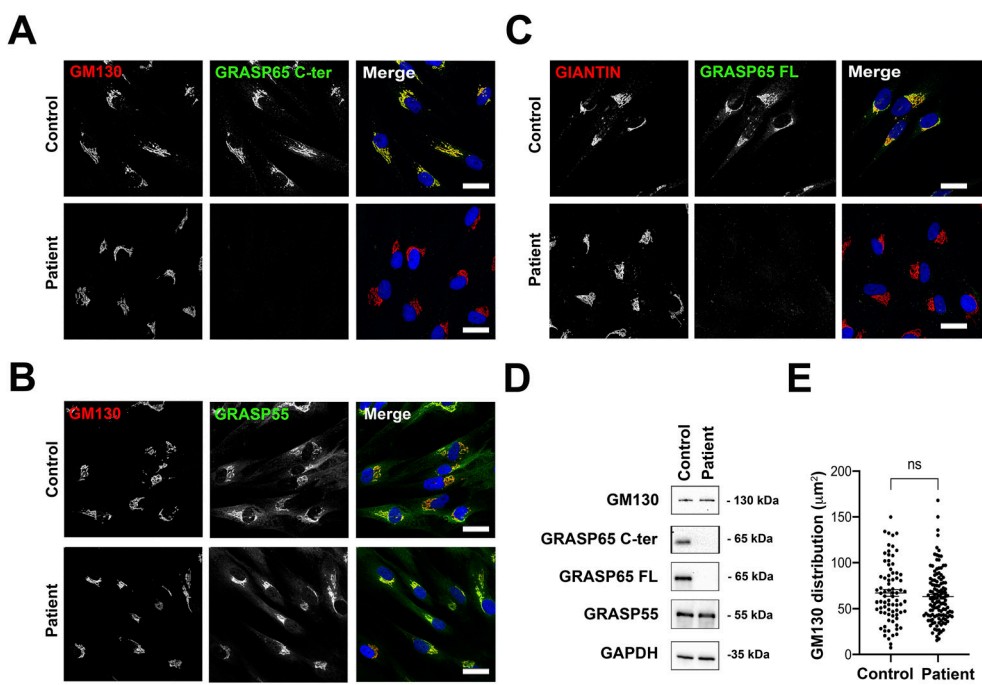

**Figure 2. Effect of the biallelic variant of *GORASP1* on GRASP65, GRASP55, and GM130 in primary human fibroblasts.**
**(A, B, C)** Immunocytochemical analysis of GRASP65 (A, C) and GRASP55 (B) in control and patient's fibroblasts labeled with GM130 (A, B) or giantin (C). Scale bars: 30 $\mu$m. **(D)** Western blot analysis of GRASPs and GM130 normalized on GAPDH in control and patient's fibroblasts. Note that GRASP65 C-ter is an antibody specific to the C terminus of the protein. GRASP65 FL corresponds to an antibody directed against the whole protein (full length). **(E)** Analysis of the overall morphology of cis-Golgi based on the area defined by GM130. Each point represents the surface area of the Golgi for a cell analyzed (n = 80–125). Data are presented as the mean ± SEM. *$P \le 0.05$, *t* test.

be linked specifically to sialylation, we labeled membranes with the lectin *Maackia amurensis*, which specifically binds sialic acid at the $\alpha 2,3$ position to galactose or N-acetylgalactosamine residues. Here too, a significant reduction in signal intensity was observed in patient fibroblasts compared with healthy controls, and the signal could be restored after GRASP65-GFP overexpression (Fig 3B). These data indicate that the loss of GRASP65 induces a reduction in membrane protein glycosylation in patient fibroblasts that affects sialylation. To determine whether these glycosylation defects could also affect intracellular proteins, we evaluated the glycosylation profile of three highly glycosylated glycoproteins, LAMP-1, LAMP-2, and TGN46, using Western blotting. In all three cases, hypo-glycosylated isoforms were more abundant in the patient than in the control, inducing greater mobility of LAMP-1, LAMP-2, and TGN46 proteins (Fig 3C). These data indicate that the loss of GRASP65 impacts glycosylation both at the cell surface and intracellularly. To further determine whether glycosylation anomalies could be detected in vivo in circulating proteins, analyses were carried out on the patient's serum. Although no relevant difference could be detected in total serum N-glycan profiles (Fig S3) nor in the N-glycosylation profile of transferrin (data not shown), slight hyposialylation of O-glycosylated apolipoprotein C-III was revealed in 2-dimensional electrophoresis (Fig 3D). Altogether, these results indicate that the loss of GRASP65 results in glycosylation defects in humans.

### GRASP65 loss induces mitotic defects in patient's fibroblasts

As several in vitro studies have suggested a role of GRASP65 in spindle dynamics and cell cycle progression, we then sought to determine whether the loss-of-function variant identified in our patient could result in mitosis abnormalities. To enrich primary cultures in mitoses, we first synchronized control and patient's

fibroblasts in the S phase with thymidine treatment and release, followed by a mitotic block using nocodazole. Staining with antibodies to $\alpha$-tubulin and pericentrin revealed that both control and patient's cells contained predominantly bipolar spindles with aligned chromosomes in the metaphase plate. Multipolar divisions were detected in a few cases in the patient's mitoses, but a significant number of metaphases showed misaligned or polar chromosomes close to and often behind the spindle pole (Fig 4A). We then wondered whether these mitotic defects could have an impact on cell cycle progression. To test this, we labeled cells with Ki67 and PH3 and assessed the proliferation index (Ki67/DAPI) and mitotic index (PH3/DAPI). Although the proliferation index was similar between control and patient fibroblasts, the mitotic index was significantly increased in the patient, suggesting a delay during mitosis (Fig 4B). Quantification of the different phases of mitosis further indicated an overrepresentation of prometaphases and metaphases in the patient's fibroblasts (data not shown). Taken together, these data suggest that the absence of GRASP65 induces chromosome alignment defects during mitosis and disrupts cell cycle progression in human fibroblasts. To determine whether these mitotic defects could affect Golgi distribution in daughter cells, we examined Golgi reorganization in cells at the end of telophase. The area of the GM130 signal and the number of objects were quantified, but revealed no significant difference between control and patient fibroblasts, suggesting that mitotic defects do not significantly affect Golgi reorganization in patient cells (Fig 4C).

### Glycosylation and mitotic defects are recapitulated after CRISPR/Cas9-induced depletion of GRASP65 in RPE cells

To validate that the glycosylation and mitotic defects observed are indeed the consequence of the variant identified in the patient, we

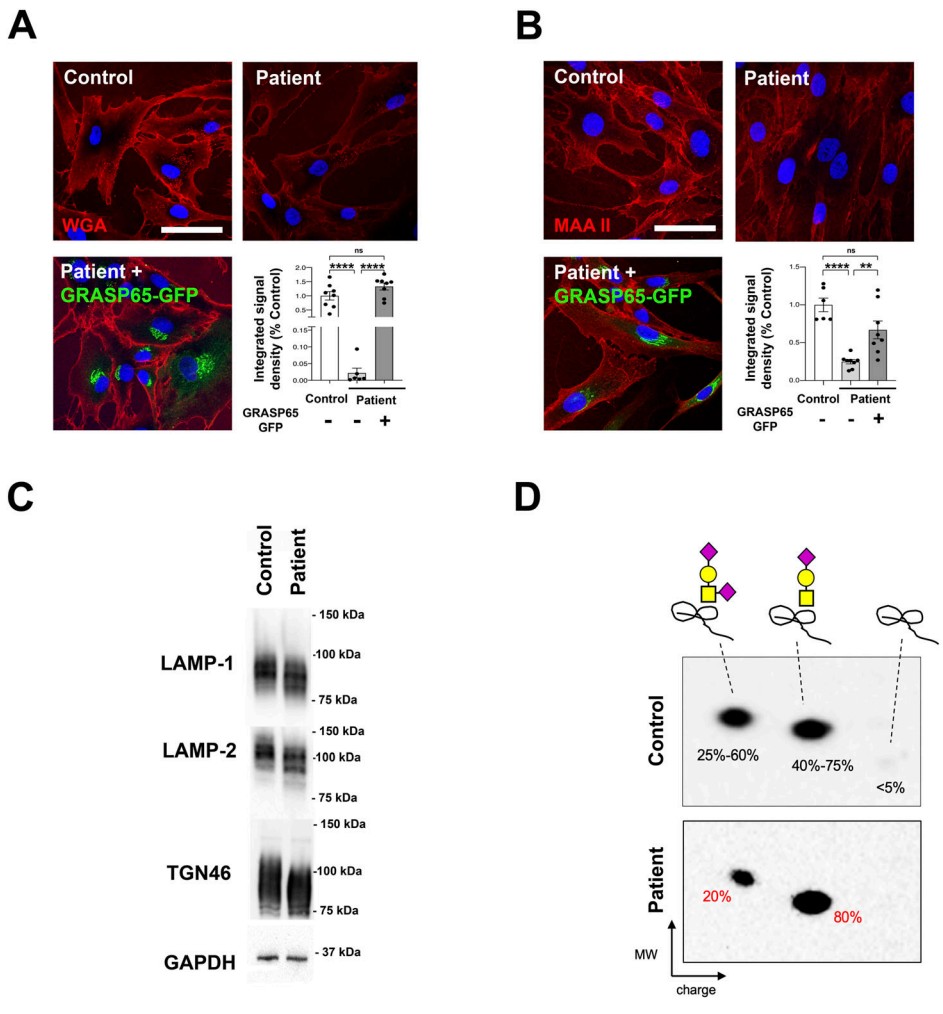

**Figure 3. Glycosylation abnormalities in primary fibroblasts and patient serum.**
**(A)** Surface staining of fluorophore-conjugated lectin WGA on non-permeabilized primary fibroblasts showing the reduced glycosylation of surface membranes in patient's cells, which is rescued upon the re-expression of GRASP65-GFP. Scale bar: 50 μm. Quantification is presented as a percentage of the mean integrated signal density of control values ± SEM. *P ≤ 0.05, one-way ANOVA. **(B)** Surface staining of fluorophore-conjugated lectin MAAII on non-permeabilized primary fibroblasts showing the reduced glycosylation of surface membranes in patient's cells, which is rescued upon the re-expression of GRASP65-GFP. Scale bar: 50 μm. Quantification is presented as a percentage of the mean integrated signal density of control values ± SEM. *P ≤ 0.05, one-way ANOVA. **(C)** Western blot analysis of LAMP-1, LAMP-2, and TGN46 normalized on GAPDH in control and patient's fibroblasts showing a shift of the bands in patient's fibroblasts. **(D)** Two-dimensional electrophoresis of the mucin core1 O-glycosylated apolipoprotein C-III (apoC-III) showing a reduced ratio of sialylation in the patient's sample. Yellow square = N-acetylgalactosamine; yellow circle = galactose; purple diamond = sialic acid.

created a similar biallelic mutation in *GORASP1* in RPE cells by CRISPR/Cas9. As in the patient's deletion, the mutation generated causes a similar frameshift in the homozygous state and the occurrence of the same premature stop codon (mutant 1, Fig S2A). As with fibroblasts, no significant difference was observed in *GORASP1* expression levels between WT RPE cells and mutant 1, indicating that the mutation exerts no effect at the transcriptional level (Fig S2B). Similarly, mutant RPE cells no longer expressed the GRASP65 protein as observed by immunofluorescence (Fig 5A) and Western blot (Fig 5B). As in the patient's cells, the overall Golgi structure was unaffected, as demonstrated by the distribution of giantin labeling (Figs 5C and S4A) and the Golgi reconstitution assay after treatment/washout with brefeldin A (Fig 5D). As expected, labeling of membrane proteins with WGA, MAA, and SNA lectins also showed a significant difference between WT and mutant RPE (Fig S4F) and an increase in TGN46 mobility (data not shown). Analysis of RPE mitoses also revealed numerous polar chromosomes and a few misaligned metaphases, but no multipolar divisions were detected (Fig 5E). As in the case of the patient's fibroblasts, mutant RPE cells showed an excess of prometaphases and metaphases at the expense of anaphases and telophases (Fig 5E), suggesting a delay in

cell cycle progression. This observation was also confirmed by the measurement of a significantly higher mitotic index in the mutant cells (Fig S4D). Of note, another mutant generated in RPE cells during CRISPR/Cas9 and that affects the same exon but eight codons downstream (mutant 2, Fig S2A) proved capable of producing a truncated GRASP65 located at the Golgi (Fig S4A) and detectable by Western blot, unlike the mutation that mimics that of the patient (Fig S4B). Interestingly, mutant 2 showed no reduction in lectin labeling of plasma membrane proteins (Fig S4F) nor in the migration profile of LAMP or TGN46 proteins (data not shown). Conversely, mitotic spindle abnormalities were much more numerous in this mutant than in mutant 1, with, in particular, a significant number of multipolar divisions resembling those previously described in HeLa cells in which GRASP65 was down-regulated by siRNA (Fig S4C). To determine whether these abnormalities could impact cell growth, RPE cells were seeded simultaneously and counted at different time points to assess the growth rate of WT versus mutant RPE cells. Remarkably, mutant 1 showed no significant difference in growth compared with WT cells, whereas mutant 2 grew obviously slower (Fig S4D and E). Altogether, these data indicate that the variant identified in the patient

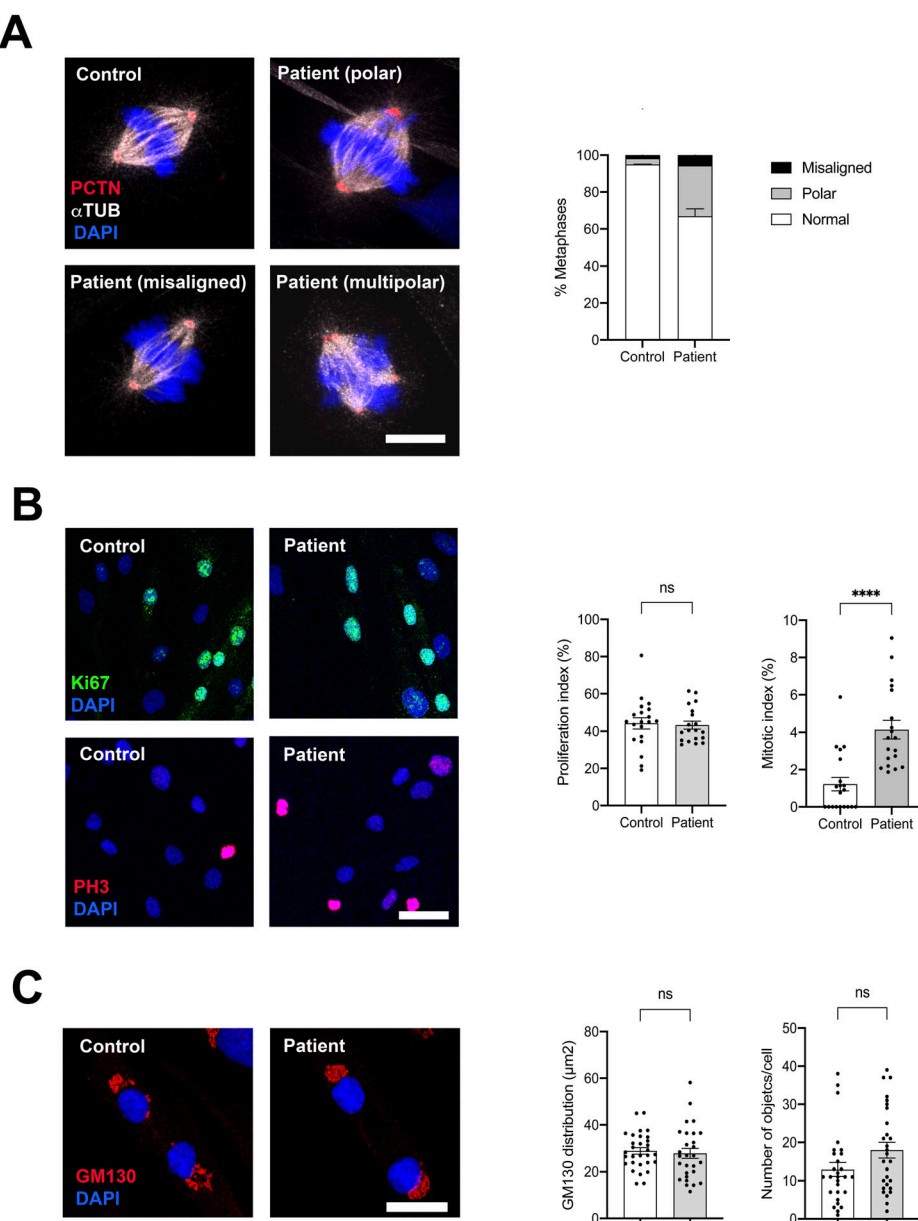

**Figure 4. Mitosis abnormalities in the patient's primary fibroblasts.**
**(A)** Immunocytochemical analysis of metaphases in control and patient's fibroblasts labeled with pericentrin and α-tubulin showing the overrepresentation of misaligned and polar spindles in the patient's cells. Data are presented as the mean ± SEM (n = 100 cells). Scale bar: 10 μm. **(B)** Immunocytochemical analysis of cycling versus mitotic cells in control and patient's fibroblasts labeled with Ki67 and phospho-histone H3 (PH3) showing a similar proliferation index (Ki67/DAPI) and an increased mitotic index (PH3/DAPI) in patient's cells as compared to the control. Data are presented as the mean ± SEM. *P ≤ 0.05, *t* test. Scale bar: 50 μm.
**(C)** Immunocytochemical analysis of telophases in control and patient's fibroblasts labeled with GM130 showing the homogeneous distribution of Golgi stacks in control and the patient daughter cells. Data are presented as the mean ± SEM. Scale bar: 20 μm.

specifically affects a region necessary for the stability of the GRASP65 protein and suggest that the consequences of a residual truncated protein on the mitotic spindle and cell growth are more deleterious than the total absence of GRASP65.

## Discussion

In this study, we report the first biallelic human variant of *GORASP1*, leading to a complete absence of the GRASP65 protein, and show that this loss of function is associated with a neurodevelopmental disorder with white matter abnormalities, neurosensory defects leading to progressive deafness and high myopia with vitreoretinal degeneration, neuromuscular contractures, truncal obesity, and

mild skeletal abnormalities. Although the diversity of tissues and organs involved can easily be explained by the ubiquitous nature of GRASP65 and the wide range of cellular functions performed by the Golgi apparatus, this complex phenotype does not correspond to any other previously described syndrome. However, it shares several characteristics with other developmental Golgipathies (Rasika et al, 2018). Neurodevelopmental anomalies, for example, and skeletal development defects are reported in a high number of cases (Passemard et al, 2017; El Ghouzzi & Boncompain, 2022). Although white matter abnormalities have been more rarely reported in Golgipathies, it is interesting to note that more diffuse and symmetrical abnormalities, as in the patient described here and suggestive of leukodystrophy, are often observed in association with defects in glycosylation pathways (Paprocka, 2023). Other

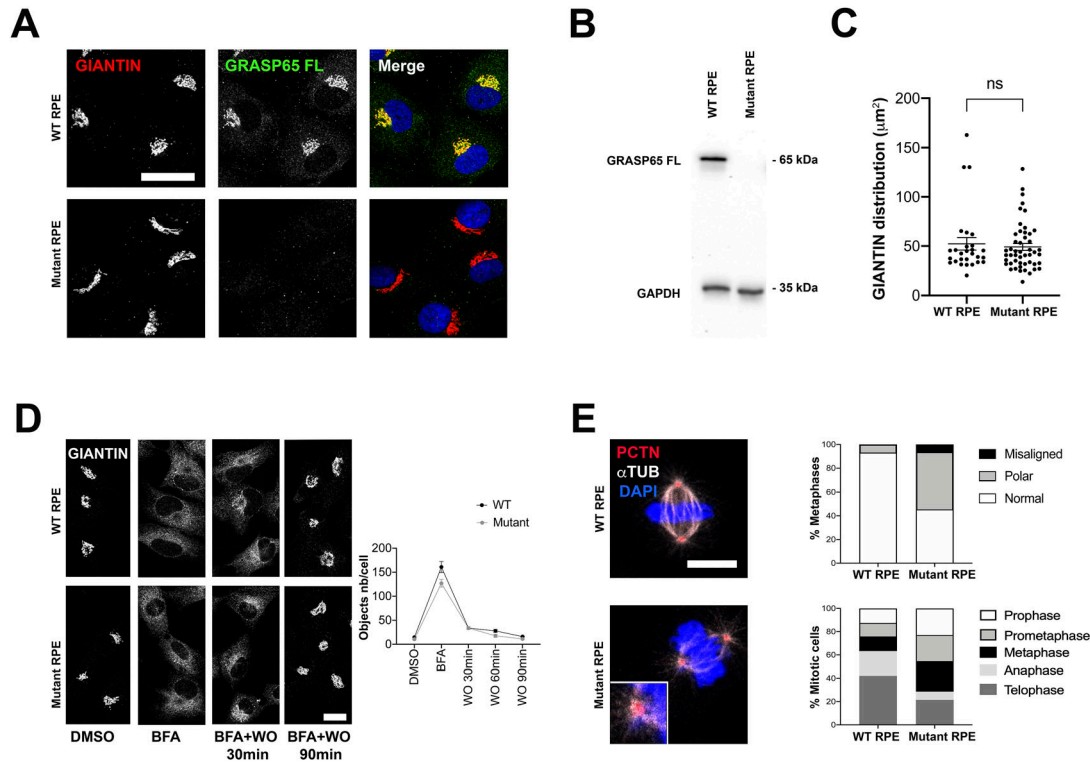

**Figure 5. Effect of loss of GRASP65 function on Golgi structure and mitosis in RPE cells.**
**(A)** Immunocytochemical analysis of GRASP65 in WT versus mutant RPE cells labeled with giantin. Scale bars: 30 $\mu$m. **(B)** Western blot analysis of GRASP65 normalized on GAPDH in WT versus mutant RPE cells. **(C)** Analysis of the overall morphology of cis-Golgi based on the area defined by giantin. Each point represents the surface area of the Golgi for a cell analyzed. Data are presented as the mean ± SEM. *$P \leq 0.05$, $t$ test. **(D)** Kinetics of Golgi reconstitution after brefeldin A (BFA) treatment and washout (WO) at different time points. Golgi structure reconstitution was quantified by evaluating the number of objects present in each cell based on the giantin signal. Scale bar: 20 $\mu$m. Data are presented as the mean ± SEM. *$P \leq 0.05$, two-way ANOVA. **(E)** Immunocytochemical analysis of metaphases in WT versus mutant RPE cells labeled with pericentrin and $\alpha$-tubulin showing the overrepresentation of misaligned and polar spindles in the patient's cells, as well as the increased ratio of prophases, prometaphases, and metaphases. Scale bar: 10 $\mu$m.

patient symptoms such as dysmorphic features, early truncal obesity, severe myopia, and learning difficulties are also found in Cohen's syndrome, a Golgipathy because of loss-of-function variants of the Golgi protein VPS13B (Güneş et al, 2023; Vacca et al, 2024). Early lattice vitreoretinal degeneration associated with neurosensorial deafness, mild joint laxity, and epiphyseal abnormalities, on the contrary, is suggestive of Stickler syndrome. Autosomal dominant or recessive transmission of Stickler syndrome is related, in the large majority of cases, to collagen genes (Acke & De Leenheer, 2022). However, a non-collagen gene, *LRP2*, encoding the low-density lipoprotein receptor and affecting endocytic uptake and clearance of morphogenic proteins is involved in rarer cases (Nixon et al, 2022). Muscular dystrophies have also been described in some Golgipathies, such as those linked to mutations in TRAPPC11 (Bögershausen et al, 2013; Corona-Rivera et al, 2024) or even GM130, the *cis*-Golgi protein that enables recruitment of GRASP65 to the Golgi membrane (Shamseldin et al, 2016; Kotecha et al, 2021). However, hypertrophic muscles without muscle weakness and elevated CPK levels, and even with improved gait as observed in our patient, are unusual for a muscular dystrophy process. Interestingly, mutations in genes involved in glycosylation pathways have been reported in patients with a form of limb–girdle

congenital myasthenic syndromes (LG-CMS) with mainly muscle weakness and fatigability similar to that in our patient (Bauché et al, 2017).

At the cellular level, we show that loss of human GRASP65 does not induce major structural abnormalities of the Golgi apparatus, but leads to defects in glycosylation of membrane and intracellular proteins in primary fibroblasts associated with a slight sialylation defect in serum apolipoprotein C-III. We also identified mitotic defects suggesting a delay in cell cycle progression without significantly impeding cell growth. RPE cells carrying a similar biallelic frameshift mutation confirm that these defects in glycosylation and mitotic regulation result from the absence of GRASP65.

GRASP65 was initially identified as a key factor enabling stack organization of the Golgi apparatus in a cell-free in vitro system (Barr et al, 1997), and cellular models of GRASP65 down-regulation by microinjection of antibodies (Wang et al, 2003) or siRNA-mediated depletion (Tang et al, 2010) have resulted in impairments in Golgi stack formation. However, clearer models of complete invalidation either by CRISPR/Cas9-induced mutations (Bekier et al, 2017) or by rapid chromophore-assisted light inactivation (Jarvela & Linstedt, 2014) have shown that GRASP65 is dispensable for stack formation and that its role is limited to tethering analogous cisternae to ensure

*cis*-Golgi compartmentalization and proper processing. These data are in agreement with our results, which reveal no difference in the overall organization of the Golgi apparatus and suggest that the consequences for Golgi integrity of an absence of GRASP65 versus residual or partial activity are completely different. *GORASP1* knockout mice published by the C. Rabouille's group confirm that the Golgi is still capable of forming stacks and self-organizes in ribbon in vivo in the absence of GRASP65, but instead shows discrete lateral disconnections of stacks restricted to the *cis*-region (Veenendaal et al, 2014). This could explain why no major phenotype was identified in the KO mouse. Interestingly, the organs affected in our patient do not appear to have been analyzed in the KO mouse model and it would be interesting to know whether some mice have white matter defects or discrete neurosensory signs. In any case, as in the mouse model, our data indicate that loss of GRASP65 does not result in compensation of GM130 or GRASP55, whose protein levels appear to be stable in patient's cells as in the RPE model. On the contrary, the patient's fibroblasts show defects in the glycosylation of membrane and intracellular proteins, which could explain the various symptoms observed in this individual. Defects in the maturation of protein-bound oligosaccharides in the Golgi apparatus, leading to the accumulation of intermediate N-glycan motifs, are known to cause congenital disorders of glycosylation, rare inherited diseases sharing variable clinical symptoms and affecting different organs such as those observed in our patient (Francisco et al, 2023; Raynor et al, 2024). It seems clear that GRASP65 loss perturbs protein glycosylation in a number of tissues. In line with this, the mouse model shows a strong reduction in membrane labeling of the lectin GSII (Veenendaal et al, 2014). Hyposialylation of apolipoprotein C-III like that found in the serum of the *GORASP1* patient has been reported in various forms of congenital disorders of glycosylation (Wada & Okamoto, 2021). Interestingly, terminal sialic acid addition occurs in the Golgi apparatus, and sialylation defects in glycoproteins are implicated in neurodevelopmental, neuromuscular, and leukodystrophic diseases (Fogli et al, 2012; van Karnebeek et al, 2016; Mullen et al, 2022). Additional patients with variants in *GORASP1* are required to clarify the role of glycosylation defects.

A specific role of GRASP65 has also been suggested in the regulation of cell division, in particular during the G2 phase of the cell cycle during which the Golgi ribbon needs to be fragmented into mini stacks to allow progression into mitosis (Yoshimura et al, 2005). Several studies showed that this disassembly process is in part regulated by phosphorylation of the GRASPs (Jesch et al, 2001; Wang et al, 2005; Tang et al, 2010; Xiang & Wang, 2010). In HeLa cells, siRNA-mediated depletion of GRASP65 leads to cell growth arrest associated with metaphase accumulation and multiple defects in mitotic spindles, including multipolar spindles (Sütterlin et al, 2005). In contrast, MEFs derived from GRASP65-invalidated mice showed no particular growth abnormalities compared with WT MEFs (Veenendaal et al, 2014). As in mice, we also failed to detect any effect of GRASP65 loss on fibroblasts or RPE cell growth, but mitosis analysis in mutant RPE cells and patient's fibroblasts revealed an over-abundance of prometaphases and metaphases, consistent with the mitotic delay described in GRASP65-depleted HeLa cells (Sütterlin et al, 2005). However, we did not observe severe abnormalities such as multipolar spindles, but only metaphases

with polar or misaligned chromosomes. Interestingly, the other mutant generated incidentally in RPE cells during CRISPR/Cas9 and that turns out to produce a C-terminally truncated but stable protein (mutant 2) induces numerous multipolar spindles associated with a significant reduction in cell growth. This mutant may disrupt GRASP65's ability to switch between an oligomerized and a phosphorylated promitotic state and induce a delay in mitosis entry. In line with this, microinjection experiments of the GRASP65 C-terminal domain into normal rat kidney cells have been previously shown to act as a dominant-negative form of GRASP65 preventing entry into mitosis (Sütterlin et al, 2002). Our data therefore fully support the idea, previously raised by C. Rabouille to explain the mild phenotype in mice (Veenendaal et al, 2014), that the total absence of GRASP65 is less deleterious than a stable form that cannot be properly phosphorylated, and suggest that most tissues and organs are capable of handling the mitotic defects observed in the patient or in RPE cells carrying the same mutation.

In conclusion, this work identifies the first biallelic variant of *GORASP1* in humans and demonstrates that loss of function of GRASP65 results in developmental defects that are not compensated for by GRASP55 and are likely to be less severe than variants compatible with a truncated form of GRASP65. The identification of additional genetic variants will enable to define the contours of this new Golgipathy, as well as the specific functions of GRASP65 in humans.

# Materials and Methods

### Genome sequencing

Genomic DNA was extracted from peripheral blood samples of the patient and his parents. Sequences were obtained by whole-genome sequencing of the father–mother–child trio at the SeqOIA laboratory (https://laboratoire-seqoia.fr/). After extraction, nucleic acids were analyzed and quantified using Spark (TECAN) and Fragment Analyzer (Agilent) and fragmented by sonication with LE220plus (Covaris). Fragment size selection and purification were carried out using Sera-Mag magnetic beads (GE Healthcare), and then, libraries were produced, without amplification, using NEBNext Ultra II End repair/A-tailing module & Ligation module (New England Biolabs) and quantified by qPCR (NEBNext Custom 2X Library Quant Kit Master Mix; New England Biolabs; QuantStudio 6 Flex Real-Time PCR System; Life Technologies). Sequencing was performed using paired-end reading (2 × 150 cycles), SBS Technology (Flow Cell S4, NovaSeq 6000; Illumina).

Bioinformatics analysis was performed by Processing Chains using pipeline_bcl2bam_wgs v1.0.0; pipline_trio_WGS v3.1.0; snakefile_analysis v3.1.1;pipeline_config v3.1; cluster_configur v3.2 Demultiplexing: bcl2fastq; Illumina (v2.20.0.422). Alignments were performed using BWA-MEM (v0.7.15), and the GRCh38.92.fa reference genome was used for SNV and indels <50 bp detection using Haplotype caller, GATK (v4.1.7. 0) SNV and indels <50 bp were annotated using SNPEff (v4.3t), and structural variants were detected and annotated using ClinSV (v1.0.1), WiseCondor (v1.2.4), and AnnotSV (v3.0.7).

## Cell culture, synchronization, and transfection

Human dermal fibroblasts from the patient and unaffected age-matched individuals were derived from skin biopsies and grown in DMEM (Thermo Fisher Scientific) supplemented with 15% FBS (Invitrogen), 1% GlutaMAX, penicillin–streptomycin, and 4.5% glucose, in a humidified incubator at 37°C with 5% $CO_2$. For transfection, fibroblasts were electroporated using the Neon transfection system (Invitrogen) and plated on coverslips for 24 h. For synchronization, the protocol was adapted from Matsui et al (2012). Briefly, fibroblasts were treated with 4 mM thymidine (Sigma-Aldrich) for 24 h, then washed once with PBS 1x, and released into thymidine-free fresh medium for 20 h. Cells were then incubated for 5 h with 30 ng/ml of nocodazole (Sigma-Aldrich) in DMSO, then washed once with PBS 1x, and put back in fresh complete medium for 30 and 40 min.

RPE1 cells were routinely cultivated in DMEM/F-12 (Thermo Fisher Scientific), supplemented with 10% FBS (Thermo Fisher Scientific) and 1% penicillin–streptomycin in a humidified incubator at 37°C with 5% $CO_2$.

## CRISPR/Cas9 gene editing

The sgRNAs targeting exon 9 of *GORASP1* were designed using Santa Cruz Tefor software (http://crispor.gi.ucsc.edu/), verified, and purchased from Integrated DNA Technologies (https://eu.idtdna.com/). RPE1 cells were electroporated using the Neon system (Invitrogen), plated onto six-well plates for 24 h, and diluted at a rate of 1 cell/well in 96-well plates to generate individual colonies. For each colony, DNA was extracted and *GORASP1* mutations were confirmed by Sanger sequencing. Positive clones were subjected to CGH analysis to exclude off-targets.

## Immunocytochemistry

Cultures were fixed for 20 min with 4% PFA in PBS. For immunocytochemistry, the cells were permeabilized with 0.3% Triton X-100 in PBS for 5 min, washed, blocked for 1 h with 10% donkey serum in PBS, and incubated with primary antibodies in 5% donkey serum in PBS overnight at 4°C. After washing, cells were treated with fluorescent secondary antibodies made in donkey (1:1,000 dilution) for 1 h in the dark at room temperature. Primary antibodies and dilutions used were as follows: GRASP65 (NP2-02665, 1:200; Novus, and MA5-34658, 1:200; Invitrogen), GRASP55 (GORASP2, 10598-1-AP, 1:100; Proteintech), GM130 (610823, 1:100; BD Biosciences), giantin (ab80864, 1:400; Abcam), α-tubulin (MCA78G, 1:500; Serotec), pericentrin (ab28144, 1:1,000; Abcam), Ki67 (M7240, 1:100; Agilent), and PH3 (9701S, 1:100; Cell Signaling). All images were captured on a Leica SP8 confocal microscope and analyzed with ImageJ.

## BFA treatment and washout assay

To trigger Golgi fragmentation, the medium containing BFA (B7651 in DMSO; Sigma-Aldrich) at a final concentration of 5 μg/ml was added to RPE cells grown on coverslips for 90 min. For BFA washout experiments, RPE cells were treated for 90 min with BFA, washed three times with fresh medium, and incubated for an additional 30, 60, and 90 min before fixation.

## Lectin staining

For lectin staining, cells were grown on coverslips to 50–70% confluency, then fixed in 2% ice-cold PFA in PBS for 15 min, washed three times with PBS, and blocked in 1% BSA in PBS for 30 min at room temperature. After blocking, cells were incubated with 2 μg/ml fluorophore-conjugated lectins (Texas Red-WGA; Thermo Fisher Scientific, TRITC-MAAII; BioWorld, or Fluorescein-SNA; Vector Laboratories) for 10 min at room temperature followed by three 10-min PBS washes and mounted onto glass slides for confocal microscope analysis.

## Western blotting

Cell pellets were treated with ice-cold lysis buffer supplemented with protease and phosphatase inhibitors and centrifuged. Protein dosage was performed using the BCA protein assay (Thermo Fisher Scientific). Lysates were heat-denatured with 1% beta-mercaptoethanol, separated by SDS–PAGE, and transferred to PVDF membranes. Membranes were blocked with 5% skimmed milk for 1 h and coated with primary antibodies at 4°C overnight, followed by HRP-linked mouse/rabbit IgG secondary antibodies for 1 h at room temperature, before detection by ECL (Amersham) using a Syngene PXi imaging system. The signal was detected by chemiluminescence and quantified using an ImageJ Gel tool. Primary antibodies used were as follows: GRASP65 (NP2-02665, 1:1,000; Novus, and MA5-34658, 1:1,000; Invitrogen), GRASP55 (GORASP2, 10598-1-AP, 1:2,000; Proteintech), GM130 (610823, 1:1,000; BD Biosciences), LAMP-1 (9091S, 1:1,000; Cell Signaling), LAMP-2 (ab25631, 1:500; Abcam), TGN46 (APH500G, 1:1,000; Bio-Rad), GAPDH (60004-1-1g, 1:10,000; Proteintech).

## Analysis of total serum N-glycans by matrix-assisted laser desorption/ionization time-of-flight mass spectrometry (MALDI-TOFMS)

Total serum N-glycan profiles were obtained by MALDI-TOFMS after N-glycan release using peptide N-glycosidase F (PNGase F), N-glycan purification by solid-phase extraction, and permethylation as previously described (Goyallon et al, 2015; Bruneel et al, 2018). Briefly, serum samples (5 μl) were diluted in 20 mM sodium phosphate buffer (pH 7.4) containing 10 mM dithiothreitol, and resulting mixtures were heated at 95°C during 5 min for protein denaturation. After an overnight incubation with PNGase F (2U; Roche Diagnostics) at 37°C and quenching by sample acidification, de-N-glycosylated proteins were precipitated using ice-cold ethanol. Released N-glycans were then purified using porous graphitic carbon solid-phase extraction cartridges (Thermo Fisher Scientific) and subsequently permethylated before MALDI-TOFMS analysis. Mass spectra were obtained on an ultrafleXtreme mass spectrometer operated in the reflectron positive-ion mode (Bruker Daltonics) using 2,5-dihydroxybenzoic acid (10 mg/ml in 50% methanol containing 10 mM sodium acetate) as a matrix. The MALDI-TOF mass spectra were

internally calibrated and further processed using GlycoWorkbench software (Ceroni et al, 2008).

### Confocal microscopy

Immunofluorescence sections were analyzed using a Leica TCS SP8 confocal scanning system (Leica Microsystems). Eight-bit digital images were collected from a single optical plane using a 20x HC PL APO CS2 oil-immersion Leica objective (numerical aperture 0.75) or a 40x HC PL APO CS2 oil-immersion Leica objective (numerical aperture 1.30). For each optical section, double- or triple-fluorescence images were acquired in sequential mode to avoid potential contamination by linkage-specific fluorescence emission cross-talk. Settings for laser intensity, beam expander, pinhole (1 Airy unit), range property of emission window, electronic zoom, gain and offset of photomultiplicator, field format, and scanning speed were optimized initially and held constant throughout the study so that all sections were digitized under the same conditions. For WGA quantification, images were segmented, mean gray and area values were measured, and integrated density values were calculated using the Fiji distribution of ImageJ (Schindelin et al, 2012). For GM130 quantification, the counting area in $\mu m^2$ was determined for each cell by the GM130 fluorescence using a homemade macro integrated in Fiji.

### Statistics

Statistical analysis was performed using GraphPad Prism 8.0 (GraphPad Software). Mean values for the experimental groups were calculated and statistically analyzed by one-way ANOVA with Sidak's post hoc test for multiple comparisons in the same dataset or by two-tailed unpaired $t$ test for single comparisons with the control. The values of $P < 0.05$ were considered statistically significant. All data in the text and legends are presented as the mean ± SEM.

### 2D electrophoresis

Two-dimensional electrophoresis of the mucin core1 O-glycosylated apolipoprotein C-III (apoC-III) has been performed as previously described (Yen-Nicolaÿ et al, 2015) on 1 $\mu l$ of serum. Briefly, isoelectric focusing (first dimension) was performed on ZOOM Strip (pH 4–7) and SDS–PAGE (second dimension) was performed on a NuPAGE Bis-Tris gel (4–12%) (Thermo Fisher Scientific). After transfer onto nitrocellulose, membranes were incubated in TTBS/5% milk with rabbit anti-human apoC-III (1/5,000 vol/vol; BioDesign International) and then with a HRP-tagged secondary antibody (1:5,000 vol/vol; GE Healthcare). Revelation was conducted using Clarity ECL reagent, and 2-DE profiles were acquired on an XRS ChemiDoc camera (Bio-Rad).

## Data Availability

The data supporting the conclusions of this article are included within the article and its additional files.

### Ethics statement

Signed informed consent forms were obtained from the patient, siblings, and both parents who participated in this study. For whole-genome analysis, documents signed have been approved by the French National Ethic Committee (SeqOIA Platform, PFMG 2025; https://pfmg2025.aviesan.fr/en/information-notices/). Additional consent forms were signed by the parents for the siblings included in the familial segregation analysis.

For research purposes, the patient and the parents have accepted to participate in the LEUCOEPIMAR project in order to improve the epidemiology, the natural history, the molecular defects, and the physiopathology of leukodystrophies. This research has been approved by the Ethic committee (Comité de Protection des Personnes Sud-Est VI N° AU788), the Sanitary Security French Agency (Agence Française de Sécurité Sanitaire des Produits de Santé, AFSSAPS, N° B90298-60), and the Informatics French Commission for the data collection in the Leuko database (Commission National Informatique et Liberté, CNIL, N° 1406552).

## Supplementary Information

## Acknowledgements

We would like to acknowledge the patient and his family who have entrusted us with the care of their child. We are grateful to Cécile Martel for invaluable support. We would also like to thank the biobank of the biochemistry department of the Kremlin-Bicêtre University Hospital (Élise Lebigot) for generating and supplying us with the patient's skin fibroblasts. This work was supported by the CRMR LEUKOFRANCE financial endowment (Brain Team DGOS, plan maladies rares 2018–2024), the Nancy & Jean Pierre Boespflug Foundation for Myopathic Research, the Institut National de la Santé et la Recherche Médicale (INSERM), the Centre National de la Recherche Scientifique (CNRS), the Paris Cité University (Stratex-Idex-2023-028 EMERGENCE [ACOMEN project]) and by research grants from the French government managed by the Agence Nationale de la Recherche (ANR-16-CE16-0024 [MicroGol project], ANR-22-CE16-0008 [MiCMac project]) and as part of the France 2030 program, under the reference ANR-23-IAHU-0010.

### Author Contributions

S Lebon: formal analysis, investigation, methodology, and writing—review and editing.
A Bruneel: formal analysis, investigation, and writing—review and editing.
S Drunat: formal analysis, investigation, and writing—review and editing.
A Albert: formal analysis.
Z Csaba: formal analysis, investigation, and methodology.
M Elmaleh: investigation.
A Ntorkou: investigation.
Y Ténier: investigation.
F Fenaille: investigation.

P Gressens: conceptualization, funding acquisition, and writing—review and editing.
S Passemard: investigation, methodology, and writing—review and editing.
O Boespflug-Tanguy: conceptualization, funding acquisition, project administration, and writing—review and editing.
I Dorboz: conceptualization, funding acquisition, investigation, project administration, and writing—original draft, review, and editing.
V El Ghouzzi: conceptualization, supervision, funding acquisition, investigation, methodology, project administration, and writing—original draft, review, and editing.

## Conflict of Interest Statement

The authors declare that they have no conflict of interest.

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
