## [Reviewer comments · Life Science Alliance]

Life Science Alliance

A biallelic variant in GORASP1 causes a novel Golgipathy with glycosylation and mitotic defects

Sophie Lebon, Arnaud Bruneel, Severine Drunat, Alexandra Albert, Zsolt Csaba, Monique Elmaleh, Alexandra Ntorkou, Yann Ténier, François Fenaille, Pierre GRESSENS, Sandrine Passemard, Odile Boespflug-Tanguy, Imen Dorboz, and Vincent El Ghouzzi

DOI: <https://doi.org/10.26508/lsa.202403065>

Corresponding author(s): Vincent El Ghouzzi, Inserm and Imen Dorboz, AP-HP

Review Timeline:

Submission Date:	2024-09-27
Editorial Decision:	2024-11-07
Revision Received:	2025-01-27
Editorial Decision:	2025-01-29
Revision Received:	2025-01-29
Accepted:	2025-01-30

Transaction Report:

November 7, 2024

Re: Life Science Alliance manuscript #LSA-2024-03065-T

Dr. Vincent El Ghouzzi
INSERM
NeuroDiderot
Hopital Robert Debre
48 Blvd Serurier
Paris, Paris 75019
France

Dear Dr. El Ghouzzi,

Thank you for submitting your manuscript entitled "A biallelic variant in GORASP1 causes a novel Golgipathy with glycosylation and mitotic defects" to Life Science Alliance. The manuscript was assessed by expert reviewers, whose comments are appended to this letter. We invite you to submit a revised manuscript addressing the Reviewer comments.

Thank you for this interesting contribution to Life Science Alliance. We are looking forward to receiving your revised manuscript.

Sincerely,

B. MANUSCRIPT ORGANIZATION AND FORMATTING:

Reviewer #1 (Comments to the Authors (Required)):

The paper of Lebon et al., describes a novel human glycosylation disease resulting from a defect in GORASP1. I found the paper well written and scientifically very interesting as this is the first human disease associated to GRASP defects. Some parts of the paper are however too descriptive to be accepted as it is. Please find my main comments that the authors would need to take into account to have an acceptable revised version of their manuscript.

Figure 2: I'm quite surprised that the Golgi is not found dilated. Have the authors tried to destructure the Golgi with either nocodazole/ BFA to follow its reorganization? The dynamic within the Golgi should be also affected. This has not been investigated. I fully recognize that such experiments are difficult to perform in patients' fibroblasts but the authors could use RPE cells.

Figure 3: This is certainly one of the most important messages of the paper. I recognize that glycosylation is overall affected but at this level it is of utmost importance to deeply analyse N-glycosylation status in patients' cells by mass spectrometry to understand the link between this defect, Golgipathy and glycosylation.

Authors use WGA to confirm their glycosylation defect. I would advice them to repeat the experiments with SNA and MAA that are more specific lectins for sialylation.

I'm also very surprised that the authors detect an APOCIII glycosylation defect and none in transferrin. The sialylation that occurs on O-glycans is in alpha 2,3 while for N-glycans this is mainly alpha 2,6. The deficiency maybe only affects the alpha 2,3 sialylation. The use of the two lectins SNA and MAA should answer this question but this is why the N-glycosylation status in patients' cells by mass spectrometry is important. This could also be done in RPE mutant cells used in the second part of the paper.

Figure 4 : The mitotic phenotype observed in patients' cells is very intriguing but purely descriptive. I would honestly keep this part away of the current version paper to reinforce the cellular trafficking/ Golgi organization and Glycosylation part of the paper. If the authors want to keep it in their version, it would be essential to show how the Golgi looks like in mitotic patients' cells and follow the reorganization of the Golgi during telophase and cytokinesis. It is possible that Golgi fragmentation remains during these cell cycle phases.

Figure 5 and Figure 4 need to be together and the confirmation that RPE cells have a glycosylation defect in supp data.

Reviewer #2 (Comments to the Authors (Required)):

Short summary

1.This paper reports the first human case of Golgipathy. These patients display neurological and skeletal issues. This is due to very specific truncation in the mRNA encoding the Golgi rotein GRASP65.

2.This mutation results in a truncation of grasp65 mRNA but absence of a a truncated protein, as shown using a anti FL antibody as well as an anti C-ter antibody by WB.

confocal microscopy also does not detect the truncated protein.

The known GRASP65 associated proteins GM130 and GRASP55 are intact in in term of level. GM130 and giantin localisation appears typical. What about GRASP55?

3.Fibroblasts from these patients show two cell biological phenotypes:

-A severe glycosylation deficit, especially in terminal glycosylation detected by WGA

- An increase of bipolar cells indicative of a delay in mitosis progression, and more mitotic defects.

4.This mutation/truncation has been recapitulated in RPE cells using CRISPR, resulting to the same cell biological phenotypes,

both rescued by reintroducing full length GRASP65-GFP.

This is in line with the studies on GRASP 65 depletion and KO in cells and in animals by several methods.

Smartly, the authors have not speculated on the Golgi structure, such as fragmentation or stacking, which would not be possible to determine using fluorescence microscopy as they do in the paper.

5. A second mutation producing also a truncated GRASP65 mRNA eight codon longer than truncation 1, has been identified in another patient exhibiting with similar developmental defects.

However, this leads to the production of truncated protein to level similar as WT, detected by WB.

6. Puzzling is that truncation 2 leads to more mitotic defects but less glycosylation defects.

It appears that a truncated GRASP65 is more detrimental than not having the full length protein. DN effect.

Points to address

Related to 1.

What does the full length antibody detect is not clear. Is the presence of the N-ter of the protein completely ruled out?

Could this be addressed? At least in RPE cells by providing GRASP65-Nter (Or version of it) instead of FL GRASP65.

Related to 1.

-The general question is whether those cell biological defects contribute to the patient phenotype in brain and bones. Are those causal?

Admittedly, in the absence of possible biopsy of tissue, direct investigation of this particular issue it is not possible to determine. Even if those defects are present in the affected tissue (mitotic delay and affected glycosylation), it remains to be established as to whether they are causal.

It is possible that GRASP65 is part of a-insofar- not elucidated signaling cascade, or in cell biological pathway, such as unconventional protein secretion, which, when affected could lead to neurological and skeletal developmental problems.

Related to 5.

-What is encoded these eight codons?

-Does this mean that the truncation 1 mRNA is unstable and not translated whereas truncation 2 mRNA is stable and translate?

-How can the presence of these eight codons explain this? Not necessarily important for this study but might be important for the RNA stability (in stress assemblies) and degradation.

field. Any speculation

-Could these two truncated mRNAs be detected and possibly localized? By smFISH.

Related to 5

Where is this GRASP65 protein produced in the truncation 2 localised? This is not shown and very clearly missing.

Importance;

The manuscript is important because it is the first reporting mutations in a "structural" protein of the Golgi. It should be published

The authors should address my points, especially the localisation of truncation 2 protein.

and whether truncation 1 results in N-ter protein fragment.

Referee Cross-Comments:

I AM glad that the first reviewer agrees with the fact that it is an important paper. It reports the first disease associated to mutation in GRASP65. To me, this should guarantee its publication as it is an important information and advance in the field.

Although I understand that knowing the exact modifications in the glycosylation pattern resulting to the lack of GRASP65, I do not think it is necessary to undergo the mass spec analysis of the glycosylation defects.

it will add more to the understanding of the relationship between the cell biology and the developmental phenotype.

This analysis is has been done in cells depleted of GRASP65.

Since is no access of bone and brain cells directly from the patients, it will not add a lot.

This detailed study could come in another study after reporting this one.

I also disagree that removing the mitotic aspect of the phenotype. GRASP65 perturbations as has been shown to be associated with mitotic defects (Sutterlin et al). It has been largely ignored but it is now clear that it is part of the phenotype. It is therefore possible that this phenotype is involved in the observed developmental defects.

Point-by-point Responses to Reviewer requests

Manuscript reference: Life Science Alliance manuscript #LSA-2024-03065-T

Lebon et al, "A biallelic variant in *GORASP1* causes a novel Golgipathy with glycosylation and mitotic defects"

Dear Editor and reviewers, thank you for your excellent and constructive comments on our manuscript #LSA-2024-03065-T by Sophie Lebon et al, which will clearly help to improve it. Please find below in blue the point-by-point answers to your comments and questions.

All changes made to the text of the manuscript or in the figure legends are highlighted in red.

Reviewer #1 (Comments to the Authors (Required)):

The paper of Lebon et al., describes a novel human glycosylation disease resulting from a defect in *GORASP1*. I found the paper well written and scientifically very interesting as this is the first human disease associated to *GRASP* defects. Some parts of the paper are however too descriptive to be accepted as it is. Please find my main comments that the authors would need to take into account to have an acceptable revised version of their manuscript.

We are delighted that reviewer 1 has appreciated our manuscript. We have now addressed all his/her queries and believe that the manuscript is much improved as a result.

Figure 2: I'm quite surprised that the Golgi is not found dilated. Have the authors tried to destructure the Golgi with either nocodazole/ BFA to follow its reorganization? The dynamic within the Golgi should be also affected. This has not been investigated. I fully recognize that such experiments are difficult to perform in patients' fibroblasts but the authors could use RPE cells.

As suggested, we added a Golgi destructuring/reorganisation experiment by treating RPE cells with BFA and quantifying Golgi reformation after washout (revised Figure 5D). No significant differences suggesting Golgi fragmentation were observed in mutant RPE, confirming our initial observations in IF. This is not necessarily surprising since several studies in other cell systems have shown that *GRASP55* (which is not affected in our cells) may partly compensate for the absence of *GRASP65* and that the single KO of one of the *GRASPs* does not result in a major defect in Golgi morphology (Bekier M, Mol Biol Cell 2017, Tang D, Traffic 2010).

In addition to the BFA assay, we also quantified the distribution of the GIANTIN signal in RPE cells which we found to be no different between WT cells and mutant 1, the corresponding graph has been added in revised Figure 5C. A sentence explaining this experiment and the experiment with BFA has been added accordingly in the text on page 15. A short paragraph detailing the BFA assay has also been added in the materials and methods section, page 6.

Figure 3: This is certainly one of the most important messages of the paper. I recognize that glycosylation is overall affected but at this level it is of utmost importance to deeply analyse N-glycosylation status in patients' cells by mass spectrometry to understand the link between this defect, Golgipathy and glycosylation. Authors use WGA to confirm their glycosylation defect. I would advice them to repeat the experiments with SNA and MAA that are more specific lectins for sialylation. I'm also very surprised that the authors detect an APOCIII glycosylation defect and none in transferrin. The sialylation that occurs on O-glycans is in alpha 2,3 while for N-glycans this is mainly alpha 2,6. The deficiency maybe only affects the alpha 2,3 sialylation. The use of the two lectins SNA and MAA should answer this question but this is why the N-glycosylation status in patients' cells by mass spectrometry is important. This could also be done in RPE mutant cells used in the second part of the paper.

We would like to thank Reviewer 1 for this important comment. We were able to obtain fresh serum from the patient and carry out an N-Glycome analysis by mass spectrometry, the figure has now been added as a Supplementary Figure 4, but shows no significant difference between patient and control. It is possible that this

result depends on the type of cell or tissue examined, and it will be very important to repeat these analyses in other samples in the future.

Regarding lectins, we followed the reviewer's recommendations and carried out additional staining for MAA lectin and SNA lectin (**revised Figure 3B and revised Supplementary Figure 3F**). Interestingly, MAA and SNA staining showed a significant reduction in GRASP65-mutated cells indicating that glycosylation defects could involve sialylation specifically. Consistent with what we observed with the WGA lectin in RPE, mutant 2 showed no reduction in signal with these new lectins, confirming that the functions of the GRASP65 involved in glycosylation are intact in this truncated mutant (**revised Supplementary Figure 3F**). Sentences have been added in the results section page 13 and page 15 to mention the results obtained with the additional lectins, and the lectin used have been cited in the "lectin staining" paragraph of the material & methods section, page 6.

Figure 4 : The mitotic phenotype observed in patients' cells is very intriguing but purely descriptive. I would honestly keep this part away of the current version paper to reinforce the cellular trafficking/ Golgi organization and Glycosylation part of the paper. If the authors want to keep it in their version, it would be essential to show how the Golgi looks like in mitotic patients' cells and follow the reorganization of the Golgi during telophase and cytokinesis. It is possible that Golgi fragmentation remains during these cell cycle phases.

We agree with Reviewer 1 that although we have observed mitotic anomalies, we have not demonstrated their effect either on the cells or on the patient's phenotype. However, mitotic spindle abnormalities in the absence of GRASP65 and a role for the protein in cell cycle entry have been reported in previous studies (Sütterlin C, Cell 2002; Sütterlin C Mol Biol Cell 2005; Tang D, Traffic 2010), and we believe that these abnormalities may contribute to the phenotype of this Golgipathy. As also requested by Reviewer 2 for the same reasons, we have chosen to retain these results in the manuscript. However, we have followed the advice of Reviewer 1 by evaluating the Golgi morphology in patient cells at the end of telophase. The distribution of Golgi in these cells is not different from that in control fibroblasts, suggesting that the absence of the protein does not cause abnormal fragmentation of Golgi stacks visible in IF and that the mitotic defects observed do not lead to abnormal distribution of Golgi in daughter cells as far as can be judged in IF. We have added these data in **revised Figure 4C**. Two sentences have been added in the result section to describe this new experiment (page 15).

Figure 5 and Figure 4 need to be together and the confirmation that RPE cells have a glycosylation defect in supp data.

In view of the new experiments added in Figures 4 and 5, we have not merged these figures, but have followed Reviewer 1's recommendation by switching the glycosylation of RPE cells (initially in Figure 5C) to **revised Supplementary Figure 3F**.

Reviewer #2 (Comments to the Authors (Required)):

Short summary

1.This paper reports the first human case of Golgipathy. These patients display neurological and skeletal issues. This is due to very specific truncation in the mRNA encoding the Golgi rotein GRASP65.

2.This mutation results in a truncation of grasp65 mRNA but absence of a a truncated protein, as shown using a anti FL antibody as well as an anti C-ter antibody by WB. confocal microscopy also does not detect the truncated protein. The known GRASP65 associated proteins GM130 and GRASP55 are intact in in term of level. GM130 and giantin localisation appears typical. What about GRASP55?

As showed in Figures 2B and 2D, GRASP55 protein level and intracellular localization also appear 'normal in patient's cells

3.Fibroblasts from these patients show two cell biological phenotypes:

- A severe glycosylation deficit, especially in terminal glycosylation detected by WGA
- An increase of bipolar cells indicative of a delay in mitosis progression, and more mitotic defects.

4.This mutation/truncation has been recapitulated in RPE cells using CRISPR, resulting to the same cell biological phenotypes, both rescued by reintroducing full length GRASP65-GFP.

This is in line with the studies on GRASP 65 depletion and KO in cells and in animals by several methods. Smartingly, the authors have not speculated on the Golgi structure, such as fragmentation or stacking, which would not be possible to determine using fluorescence microscopy as they do in the paper.

5.A second mutation producing also a truncated GRASP65 mRNA eight codon longer than truncation 1, has been

identified in another patient exhibiting similar developmental defects. However, this leads to the production of truncated protein to level similar as WT, detected by WB.

6. Puzzling is that truncation 2 leads to more mitotic defects but less glycosylation defects.

It appears that a truncated GRASP65 is more detrimental than not having the full length protein. DN effect.

Points to address

Related to 1.

What does the full length antibody detect is not clear. Is the presence of the N-ter of the protein completely ruled out?

Could this be addressed? At least in RPE cells by providing GRASP65-Nter (Or version of it) instead of FL GRASP65.

We would like to thank Reviewer 2 for this comment and provide the following clarification: in order to exclude the presence of an N-terminal truncated product, the initial analyses were carried out with an anti-GRASP65 N-ter recognizing the first 215 amino acids and supplied by Proteintech (ref 10747-2-AP). However, this antibody was found to give numerous unspecific bands in addition to the expected band corresponding to GRASP65 (black arrowheads in the figure provided below). The anti-GRASP65 FL antibody (Novus, ref NBP2-02665) was produced against the whole human recombinant protein (sequence ref NP_114105) and as shown in the blot below, gave only specific signal. In addition, the fact that the anti-GRASP65 FL antibody is able to recognize the truncated mutant in the RPE (mutant 2) confirms that this antibody would recognize an N-ter fragment in patient's cells or mutant 1 if there were one. Consequently, we concluded the mutation identified in the patient does not produce any protein at all.

Related to 1.

-The general question is whether those cell biological defects contribute to the patient phenotype in brain and bones. Are those causal?

Admittedly, in the absence of possible biopsy of tissue, direct investigation of this particular issue it is not possible to determine.

Even if those defects are present in the affected tissue (mitotic delay and affected glycosylation), it remains to be established as to whether they are causal.

It is possible that GRASP65 is part of a-insofar- not elucidated signaling cascade, or in cell biological pathway, such as unconventional protein secretion, which, when affected could lead to neurological and skeletal developmental problems.

We thank Reviewer 2 for these thoughts, which we fully share, and to which we would like to add that the identification of additional patients in the future will provide some answers. It would also be very interesting to look for a potential pathological phenotype in the muscle, vertebrae and/or white matter of the *GORASP1* mouse model (as the organs examined in Veenendaal's paper (Biol Open 2014) mention organs other than those affected in our patient) to examine glycosylation and potential mitosis defects in these tissues.

Related to 5.

-What is encoded these eight codons?

As can be seen in the revised supplementary Figure 2A, the codons retained by mutant 2 encode the amino acids "DSLTSAA" but another major difference with the patient and mutant 1 is that mutant 2 then has 23 aberrant C-terminal amino acids before the new stop codon (shown in red in the figure).

-Does this mean that the truncation 1 mRNA is unstable and not translated whereas truncation 2 mRNA is stable and translate?

-How can the presence of these eight codons explain this? Not necessarily important for this study but might be important for the RNA stability (in stress assemblies) and degradation.

field. Any speculation

-Could these two truncated mRNAs be detected and possibly localized? By smFISH.

To address the issue of the stability of the GORASP1 transcript, we have performed qPCR on fibroblasts and RPE mRNA, which we have now added to the revised supplementary Figure 2B, showing that the transcripts are stable in the patient and in both RPE mutants. Actually, it is relatively common for a transcript carrying a truncating mutation near the 3' end to escape nonsense-mediated mRNA decay. The mutations therefore do not induce instability at transcriptional level, but whereas mutations in the patient and mutant 1 induce protein instability (no product visible in WBlot), mutant 2 produces a shorter protein that is clearly detectable. It is possible that the terminal sequence of the protein from the patient mutation (and RPE mutant 1) generates an unstable three-dimensional fold, whereas mutant 2 mRNA gives rise to a truncated but stable protein.

Related to 5

Where is this GRASP65 protein produced in the truncation 2 localised? This is not shown and very clearly missing.

We would like to thank Reviewer 2 for this comment. We have corrected this oversight by providing an image in the revised supplementary figure 3A which shows that the truncated protein produced by mutant 2 is localized to the Golgi with no apparent difference from the localization of the wild-type protein. As expected, the truncated protein is no longer detected by the antibody directed against the C-terminal part of GRASP65.

Importance;

The manuscript is important because it is the first reporting mutations in a "structural" protein of the Golgi. It should be published

We are delighted that the reviewer considers this work to be important and worthy of publication.

The authors should I dress my points, especially the localisation of truncation 2 protein. done and whether truncation 1 results in N-ter protein fragment. done

Referee Cross-Comments:

I AM glad that the first reviewer agrees with the fact that it is an important paper. It reports the first disease associated to mutation in GRASP65. To me, this should guarantee its publication as it is an important information and advance in the field.

Although I understand that knowing the exact modifications in the glycosylation pattern resulting to the lack of GRASP65, I do not think it is necessary to undergo the mass spec analysis of the glycosylation defects. it will add more to the understanding of the relationship between the cell biology and the developmental phenotype.

This analysis is has been done in cells depleted of GRASP65.

Since is no access of bone and brain cells directly from the patients, it will not add a lot.

We have added a N-Glycome analysis performed by mass spectrometry on the patient's serum as it was a request from Reviewer 1 (Supplementary Figure 4). This analysis does not reveal any significant difference compared with control, but we fully agree with Reviewer 2 that it would make more sense to be able to carry out this type of analyze on a more relevant tissue.

This detailed study could come in another study after reporting this one.

I also disagree that removing the mitotic aspect of the phenotype. GRAPS65 perturbations as has been shown to be associated with mitotic defects (Sutterlin et al). It has been largely ignored but it is now clear that it is part of the phenotype. It is therefore possible that this phenotype is involved in the observed developmental defects.

In view of these comments, which we share with Reviewer 2, we have decided to keep the mitotic phenotype in the manuscript. However, we have followed the advice of Reviewer 1 by evaluating the Golgi morphology in patient cells at the end of telophase. The distribution of Golgi in these cells is not different from that in control fibroblasts, suggesting that the absence of the protein does not cause abnormal fragmentation of Golgi stacks visible in IF and that the mitotic defects observed do not lead to abnormal distribution of Golgi in daughter cells as

far as can be judged in IF. We have added these data in **revised Figure 4C**. Two sentences have been added in the result section to describe this new experiment (page 15).

January 29, 2025

RE: Life Science Alliance Manuscript #LSA-2024-03065-TR

Dr. Vincent El Ghouzzi
Inserm
NeuroDiderot
Hopital Robert Debre
48 Blvd Serurier
Paris, Paris 75019
France

Dear Dr. El Ghouzzi,

Thank you for submitting your revised manuscript entitled "A biallelic variant in GORASP1 causes a novel Golgipathy with glycosylation and mitotic defects". We would be happy to publish your paper in Life Science Alliance pending final revisions necessary to meet our formatting guidelines.

- please be sure that the authorship listing and order is correct
- please add ORCID ID for the secondary corresponding author -- they should have received instructions on how to do so
- please add the Twitter/X and Bluesky handles of your host institute/organization as well as your own or/and one of the authors in our system
- please consult our manuscript preparation guidelines <https://www.life-science-alliance.org/manuscript-prep> and make sure your manuscript sections are in the correct order
- the contributions selected for Pierre GRESSENS do not qualify them for authorship. Please either update the contributions in our system and the Author Contributions section of the manuscript or let us know if the author needs to be removed (and added eventually to the acknowledgment section)

LSA now encourages authors to provide a 30-60 second video where the study is briefly explained. We will use these videos on social media to promote the published paper and the presenting author (for examples, see <https://docs.google.com/document/d/1-UWCfbE4pGcDdcgzcmiuJl2XMBJnxKYeqRvLLrLS08s/edit?usp=sharing>). Corresponding or first-authors are welcome to submit the video. Please submit only one video per manuscript. The video can be emailed to contact@life-science-alliance.org

A. FINAL FILES:

B. MANUSCRIPT ORGANIZATION AND FORMATTING:

Thank you for your attention to these final processing requirements. Please revise and format the manuscript and upload materials within 5 days.

Sincerely,

January 30, 2025

RE: Life Science Alliance Manuscript #LSA-2024-03065-TRR

Dr. Vincent El Ghouzzi
Inserm
NeuroDiderot
Hopital Robert Debre
48 Blvd Serurier
Paris, Paris 75019
France

Dear Dr. El Ghouzzi,

Thank you for submitting your Research Article entitled "A biallelic variant in GORASP1 causes a novel Golgipathy with glycosylation and mitotic defects". It is a pleasure to let you know that your manuscript is now accepted for publication in Life Science Alliance. Congratulations on this interesting work.

DISTRIBUTION OF MATERIALS:

Again, congratulations on a very nice paper. I hope you found the review process to be constructive and are pleased with how the manuscript was handled editorially. We look forward to future exciting submissions from your lab.

Sincerely,
